# Genome-Wide Association Studies for Flesh Color and Intramuscular Fat in (Duroc × Landrace × Large White) Crossbred Commercial Pigs

**DOI:** 10.3390/genes13112131

**Published:** 2022-11-16

**Authors:** Hao Li, Cineng Xu, Fanming Meng, Zekai Yao, Zhenfei Fan, Yingshan Yang, Xianglun Meng, Yuexin Zhan, Ying Sun, Fucai Ma, Jifei Yang, Ming Yang, Jie Yang, Zhenfang Wu, Gengyuan Cai, Enqin Zheng

**Affiliations:** 1College of Animal Science and National Engineering Research Center for Breeding Swine Industry, South China Agricultural University, Guangzhou 510642, China; 2State Key Laboratory of Livestock and Poultry Breeding, Guangdong Key Laboratory of Animal Breeding and Nutrition, Institute of Animal Science, Guangdong Academy of Agricultural Sciences, Guangzhou 510640, China; 3College of Animal Science and Technology, Zhongkai University of Agriculture and Engineering, Guangzhou 510225, China; 4Guangdong Provincial Key Laboratory of Agro-Animal Genomics and Molecular Breeding, South China Agricultural University, Guangzhou 510642, China; 5Yunfu Subcenter of Guangdong Laboratory for Lingnan Modern Agriculture, Yunfu 527400, China

**Keywords:** meat quality, genome-wide association studies (GWAS), intramuscular fat, flesh color, pig, single nucleotide polymorphism (SNP)

## Abstract

The intuitive impression of pork is extremely important in terms of whether consumers are enthusiastic about purchasing it. Flesh color and intramuscular fat (IMF) are indispensable indicators in meat quality assessment. In this study, we determined the flesh color and intramuscular fat at 45 min and 12 h after slaughter (45 mFC, 45 mIMF, 12 hFC, and 12 hIMF) of 1518 commercial Duroc × Landrace × Large White (DLY) pigs. We performed a single nucleotide polymorphism (SNP) genome-wide association study (GWAS) analysis with 28,066 SNPs. This experiment found that the correlation between 45 mFC and 12 hFC was 0.343. The correlation between 45 mIMF and 12 hIMF was 0.238. The heritability of the traits 45 mFC, 12 hFC, 45 mIMF, and 12 hIMF was 0.112, 0.217, 0.139, and 0.178, respectively, and we identified seven SNPs for flesh color and three SNPs for IMF. Finally, several candidate genes regulating these four traits were identified. Three candidate genes related to flesh color were provided: *SNCAIP* and *PRR16* on SSC2, *ST3GAL4* on SSC5, and *GALR1* on SSC1. A total of three candidate genes related to intramuscular fat were found, including *ABLIM3* on SSC2, *DPH5* on SSC4, and *DOCK10* on SSC15. Furthermore, GO and KEGG analysis revealed that these genes are involved in the regulation of apoptosis and are implicated in functions such as pigmentation and skeletal muscle metabolism. This study applied GWAS to analyze the scoring results of flesh color and IMF in different time periods, and it further revealed the genetic structure of flesh color and IMF traits, which may provide important genetic loci for the subsequent improvement of pig meat quality traits.

## 1. Introduction

With the rise in consumption levels, consumers are increasingly demanding high pork quality, and they are willing to pay for high quality pork [1]. Meat quality is measured by flesh color, drip loss, shear force, cooking loss, intramuscular fat content (IMF), etc. [2,3]. The marbling and flesh color of pork usually depend on IMF [4], which refers to the fat entrapped in lean meat by muscle connective tissue, and it is usually regarded as the quality standard of meats and is an important economic trait of domestic animals [5,6]. Therefore, the study of flesh color and IMF is of great significance and is influenced by genetics, nutrition, slaughtering method, and many other factors, of which genetic influence is great [7,8]. The study by Miar Y et al. [9] found that the heritability of flesh color and IMF were 0.20 and 0.23, respectively, and the heritability of flesh color and IMF was low to moderate. In recent years, many genes and single nucleotide polymorphism (SNP) loci related to meat quality have been detected using bioinformatics methods, of which genome-wide association studies (GWAS) have become a major method [10,11]. GWAS is an effective method used to analyze quantitative traits, and it has been widely used in the genetic analysis of quantitative traits in livestock and poultry [12,13]. Up until 24 August 2022, the pig quantitative trait locus (QTLs) database [14] included 70 QTLs for flesh color score (0.20% for), 890 QTLs (2.48%) for IMF, and 136 QTLs (0.38%) for marble score. The lack of these traits in pig QTL may be due to the complexity of the assay and the uncontrollable environment, which means that there are traits that still have the capacity for supplementation in this database. Salas et al. [15] found that pale soft exudative (PSE) meat was associated significantly with halothane genes, which is one of the causal genes for meat quality. Zhang et al. [16] demonstrated that the *PRKAG3* gene may have a significant impact on flesh color and pH. In addition, Wang et al. [17] found that *ZDHHC16*, *LOC102162218,* and *PCDH7* were significantly associated with IMF. Davoli et al. [18] identified *PPP3CA* as a potential candidate gene affecting IMF. Luo et al. [19] detected 36 SNPs on SSC12 using the 60k chip, and these SNPs are associated with IMF, marbling, and moisture [20,21,22]. Since previous studies have likely been mainly limited to population size or purebred status, resulting in large QTL spacing, and the causal genes have not been identified. In order to reveal the genetic mechanisms of meat quality, researchers have used the genome method to conduct in-depth analyses. Liu et al. [23] characterized the expression QTL (eQTL) in porcine skeletal muscle using RNA sequence data from 189 Duroc × Luchuan crossbred pigs, combined with genome-wide eQTL and allele specific expression (ASE) analysis that identified several new candidate genes for meat quality traits. In order to provide new genetic loci for the genetic improvement of meat quality traits and to better identify the QTL recombination of meat quality traits [24], we used a 1518 Duroc × Landrace × Large White (DLY) crossbred pig population to analyze the four traits (45 mFC, 12 hFC, 45 mIMF, and 12 hIMF) through GWAS, combined with the actual character correlation and heritability. We expect to provide new insight into the identification of candidate genes for flesh color and IMF and the molecular breeding of pigs.

## 2. Materials and Methods

### 2.1. Ethical Statement

All samples required for the experimental procedures were collected in this research according to the guidelines of Animal Care and Use by the China Agriculture and Rural Affairs Administration. The Animal Care and Use Committee of South China Agricultural University (SCAU) approved all animal experiments described in this study (2018F098).

### 2.2. Animals and Phenotypic Measurements

The experimental animals were provided by Guangdong Wens Foodstuff Group Co., Ltd. (Guangdong, China). A total of 1518 DLY pigs (757 boars and 764 sows) born from 2018 to 2019 were used for the experiment. These pigs were fattened from 3 farms and were divided into 13 batches for slaughter. The longissimus dorsi muscle between the 11th and 12th ribs was taken 45 min and 12 h after slaughter and scored for flesh color and IMF using the US NPPC (1991). The scoring was performed by three trained professionals and then averaged.

### 2.3. Genotyping and Quality Control

A total of 1518 ear tissues from the DLY pig was collected. Extraction of DNA from the ear tissues was completed using the traditional standard phenol/chloroform method, and the quality of the DNA samples was assessed using the light absorption ratio (A260/280) of a spectrophotometer (Guangzhou Shenhua Biotechnology Co., Ltd., Guangzhou, China) and agarose gel electrophoresis [25,26]. Genotyping was performed according to the method described by Ding et al. [27]. According to the manufacturer’s requirements, 1518 pigs were genotyped with the GeneSeek Porcine 50K SNP chip (Neogen, Lincoln, NE, USA) and quality filtering was completed using PLINK v1.90 [28]. Individuals with call rates of > 95% and markers with call rates of > 99%, a minor allele frequency (MAF) of > 99%, and a Hardy–Weinberg test *p*-value of > 10^−6^ were retained. In addition, all markers located on sex chromosomes and unmapped regions were excluded from analysis [29,30]. Briefly, a total of 28,066 SNPs were finally obtained for subsequent analysis after data editing. Furthermore, the heritability of the traits was estimated using GCTA v1.92 software in this study [31].

### 2.4. Single-Locus Genome-Wide Association Study

The GWAS refers to the method proposed by Zhou et al. [32] to correct the population structure, and an efficient mixed linear model (MLM) was used to analyze the effect of a single SNP on phenotypic traits to achieve a genetic algorithm [33,34,35]. GEMMA software was used to conduct correlation analysis on the meat quality traits in the study [36,37]. The four traits of meat quality were analyzed, respectively. The formula used in this experiment is as follows:y=Wα+xβ+μ+ε;
$$u\;\sim \;MVN_{n}\left(0,\lambda \tau^{-1}Κ\right),\epsilon \;\sim \;MVN_{n}\left(0,\tau^{-1}{Ι}_{n}\right)\;$$
where y is the phenotypic trait vector for all individuals, including 45 mFC, 12 hFC, 45 mIMF, and 12 hIMF. W is the correlation matrix based on the appropriate fixed effects, including the top five eigenvectors of the principal components analysis (PCA), slaughter batch, sex, farm, and liveweight; α is the vector of the corresponding coefficients, including the intercept; x is the vector of the SNP marker genotypes; β is the corresponding effective vector of the SNP markers; and μ and ε are the vectors of random effects and random errors, respectively. $\tau^{-1}$ is the variance of the random errors, λ is the ratio between the two variance components, Κ is the standard correlation matrix of the software estimation, ${Ι}_{n}$ is an identity matrix, and $MVN_{n}$ is a multivariate normal distribution. Since the Bonferroni correction is excessively stringent for false negative results, we refer instead to the false discovery rate (FDR) method [30,38,39] to limit the threshold of the GWAS results, and the formula for FDR screening is P=FDR∗X/N, where the FDR is set to 0.01, X is the number of SNPs (with *p* < 0.01 in GWAS results), and N is the total number of SNPs that meet the requirements of the flesh color and intramuscular fat score. The thresholds of 45 mFC, 12 hFC, 45 mIMF, and 12 hIMF set by the FDR are 1.048 × 10^−4^, 1.147 × 10^−4^, 9.371 × 10^−5^, and 9.691 × 10^−5^, respectively.

### 2.5. Linkage Disequilibrium and Gene Annotation

QTL mapping was used for the linkage disequilibrium (LD) between the whole genome genetic markers and a specific quantitative trait designed to determine the locus affecting the trait and its position on a chromosome [40]. Haplotype block analysis was carried out using PLINK v1.90 [28] and Haploview [41] software. According Gebriel et al. [42], the search for genes was within the SNP locus and 500 Kb upstream and downstream of the GWAS-significant SNP [43] on Sscrofa11.1 in the Ensembl database (http://asia.ensembl.org/index.html accessed on 13 October 2022). Gene function annotation was performed on the screened genes, and KOBAS 3.0 [44] was used to further perform gene ontology (GO) and Kyoto Encyclopedia of Genes and Genomes (KEGG) enrichment analysis on all of the screened genes.

## 3. Result and Discussion

### 3.1. Phenotypic Statistical Analysis

Table 1 shows the average scores of the DLY commercial pork color for 45 mFC, 12 hFC, 45 mIMF, and 12 hIMF. In our results, the phenotypes of the four traits measured were all normally distributed (Appendix A). The population stratification was properly adjusted before the GWAS analysis and the genome inflation factor λ range was 0.989–1.047 (Figure 1). As shown in Table 1, the coefficients of variation (CV) for 45 mFC and 12 hFC were 27.6% and 42.2%, and the CVs of 45 mIMF and 12 hIMF were 41.1% and 42.1%, respectively. In addition, this study found that the heritability of both flesh color and IMF at 12 h was greater than that at 45 min, and the gap in the heritability of flesh color was more obvious. In comparison with previous studies [45], we obtained different results due to different measurement methods and breeds, suggesting that there is considerable potential for genetic improvement in flesh color and IMF. The genomic heritability of flesh color and IMF score was 0.112–0.217 (low to medium heritability). The flesh color heritability was lower than the 0.35 found by Zhang et al. [46] in the study of six populations (2448 pigs were tested for slaughter performance), and the difference in heritability may be due to the location of the phenotype measurement and the time of the postmortem evaluation. According to the American Pork Committee, the average subjective score of flesh color in the store was 2.85 ± 0.79, the color score of the subjective in the laboratory was 2.74 ± 0.79, the average subjective score of IMF in the store was 2.30 ± 1.07, and the average subjective score of the IMF in the laboratory was 2.27 ± 1.02 [47], which is similar to our findings, where 45 mFC was larger than this data and the IMF was smaller than this data. As seen in Table 1, the average score of 45 mFC was 3.650 ± 1.006, which is 1.5 higher than 12 hFC (2.193 ± 0.925). The decline of the score may have been caused by fat deposition, carotenoids, myoglobin content decreases, and the formation of lipid peroxides [48,49,50]. As long-term storage after slaughter causes a drop in pH, which created a visual difference for scorers, the oxidation of the freshly slaughtered pork formed an oxide layer that was visually pink and oxidized to brown over time, resulting in lower scores [47]. The IMF scores obtained in our experiment were similar in 45 m and 12 h, indicating that the subjective score of the IMF was less affected by the environment. The heritability of 45 mFC and 12 hFC were 0.112 and 0.139, respectively, and the phenotypic correlation between them was 0.343 (*p* = 2.2 × 10^−16^). The heritability of the IMF score at 45 m and 12 h was higher than that of flesh color, reaching 0.217 and 0.178, respectively, and the correlation between them was 0.238 (*p* = 2.2 × 10^−16^). It is worth noting that the deposition of the IMF affected the flesh color score; therefore, we estimated the correlation between flesh color and IMF at the same time, and the correlation between flesh color score and IMF score at 45 min was 0.135 (*p* = 2.2 × 10^−16^) and the correlation between them at 12 h was 0.293 (*p* = 2.2 × 10^−16^). This indicates that the two traits are somewhat correlated, with a slightly higher correlation at 12 h. Luo et al. [19] showed that the average scores of flesh color and IMF were 3.31 and 2.88, respectively, and the correlation between flesh color and IMF was 0.29, which was consistent with the correlation between the two traits at 12 h in our study. In conclusion, the flesh color scores in previous studies were consistent with those of our experiment, but the IMF scores were slightly higher than ours, which may be due to breed differences and personal score rating errors.

### 3.2. Flesh Color GWAS

Flesh color is the first perception of consumers, and it is largely determined by myoglobin, hemoglobin, and Fe^2+^ [49,51,52]. Because bloodletting after slaughter can lead to a loss of hemoglobin, we selected 45 min and 12 h after slaughter to evaluate the muscle color of the same pig, and then we recorded the phenotype for the GWAS. According to the method mentioned above, we used the FDR method for 45 mFC and 12 hFC for the statistics of the dominant loci. All the scores of 45 mFC and 12 hFC were statistically analyzed. Within the range set by the FDR, the number of SNPs in 45 mMC and 12 hMC was four and three, respectively, and the respective *p*-values were 1.048 × 10^−4^ and 1.147 × 10^−4^. A total of seven SNPs was detected on the flesh color trait, of which four SNPs on 45 mMC were located in the 2 Mb (123.9–125.9 Mb) region on SSC2. We used haplotype software for the LD analysis of the SNPs, and there was a 79 Kb haplotype block near the top SNP (Wu_10.2_2_13124958, Figure 1e). The gene closest to the top SNP (WU_10.2_2_131124958) was *S*ynuclein α interacting protein (*SNCAIP*). DRGA0003514 on SSC2 was located at 201 Kb near *P*roline rich 16 (*PRR16*). *PRR16* has been found to be able to control the size of mammalian cells, and the proline content was relatively high. The size of the cells was related to the level of intracellular proteins, and the proteins were related to flavor. Therefore, *PRR16* can be considered as a gene related to meat quality [53]. There were three significant SNPs identified in 12 hFC (1.147 × 10^−4^), of which the top SNP (ASGA0004802) was located on SSC1, with *G*alanin receptor 1 (*GALR1*) at 246 Kb away from it. The *G*AL family is a *G* protein-coupled receptor which affects the contraction and inflammation of myometrium in pigs, and it is also a glycine receptor. *G*lycine is one of the main flavor amino acids in meat [54]. The remaining SNP (WU_10.2_9_59299494) was located at SSC9, which was located at *S*T3 β-galactoside α-2,3-sialyltransferase 4 (*ST3GAL4*) within the interval. *ST3GAL4* is involved in the synthesis of terminal N-glycans in various cells and tissues and is associated with sialic acid. In mammals, sialic acid is usually used as the terminal monosaccharide of glycans in glycoproteins and glycolipids [55]. The gene information and distance near these SNPs are listed in Table 2. As early as 1998, Andersson et al. [56] used microsatellite markers to map QTLs for flesh color traits in European wild boars and large white pigs, and they mapped the QTLs affecting pork color on SSC2. In 2001, Malek et al. [57] located QTLs in 144.2–144.5 Mb on this chromosome, and a QTL interval affecting flesh color scores was determined by using the least square regression interval and molecular genome scanning. Subsequently, Rohrer et al. [58] performed genome scanning on the loci affecting meat quality traits in a Duroc × Landrace F2 population, and they found that the region of 151.0–155.0 Mb on SSC2 was related to flesh color scores, but we found that these studies did not overlap with our findings when we analyzed a large number of DLY pigs, and so we believe this is a new QTL interval that affects flesh color scores. The interval 123–127 Mb affecting the flesh color scores obtained can be used as a supplement to the QTL affecting pork meat quality traits. We found that the flesh color error rate measured at 45 min was smaller and the score was higher, and two genes (*SNCAIP* and *PRR16*) related to meat traits were found on SSC2.

### 3.3. IMF GWAS

The IMF score is based on the marble score, which shows a significant positive correlation [59] and is also tightly related to the breed [60]. The candidate genes for IMF include *ADRP* [61] and *CAST* [62]. Stewart et al. [63] used the relationship between marbling score and IMF to predict the palatability of beef to consumers, and IMF content and marbling were found to have remarkably similar precision. According to our previous calculation method, three SNPs in 12 hIMF were identified as significant loci (Table 2). However, no significant SNPs associated with IMF score measured at 45 min were found. It may be that some SNP sites had been deleted because the labeling density and conditions were too strict. Subsequently, new SNPs on IMF can be selected by increasing the labeling density or selecting a looser quality control condition [43]. Although only three SNPs were screened on 12 hIMF, both SSC2 (H3GA0054148) and SSC15 (WU_10.2_15_140224592) reached a loose genome-wide significant threshold (1/28066). An SNP on SSC2 was located on *A*ctin binding LIM protein 3 (*ABLIM3*), which was identified as a candidate gene for IMF by GWAS in Chinese Lulai black pigs [64] associated with fat deposits. An SNP on SSC15 was found at 1.3 kb downstream of *DOCK10* in a GWAS of backfat thickness in Italian white pigs, and Fontanesi et al. [65] identified this gene as a candidate gene affecting subcutaneous fat at 126.5 Mb on SSC15, which is consistent with our findings. The *DPH5* gene was found on SSC4 and the *DPH5* protein is produced in Escherichia coli, and studies have shown that the elongation factor 2 of *DPH5* gene deletion mutants has weak ADP-ribose receptor activity [66] and may be related to the metabolism of the animal’s body. In addition, the gene *GPC6* found by Ding et al. [27] using GWAS that affected IMF deposition was also validated, but it was not found in our study. We analyzed the interval (126–127 Mb) of the top SNP (WU_10.2_15_140224592) on SSC15 and found that the linkage degree between WU_10.2_15_140224592 and MARC0011865 was relatively high (*r*^2^ = 0.50). The top SNP showed moderate correlations with the remaining two SNPs, indicating that our research results on the IMF score have certain reference value.

### 3.4. Pathway Enrichment Analysis

We performed gene function enrichment analysis on the enriched genes using KOBAS 3.0 and enriched the four and six pathways on KEGG and GO, respectively (Appendix A). On the KEGG enrichment pathway, *ADRB2* and *HTR4* were jointly enriched on the neuroactive ligand-receptor interaction, calcium signaling pathway, and cAMP signaling pathway, and two genes—*CSNK1A1* and *CUL3*—were enriched on the hedgehog signaling pathway. The Ca^2+^ signaling pathway is controlled by the increase and decrease in Ca^2+^ levels in the cytoplasm, and the regulation of Ca^2+^ levels in the cytoplasm depends on Ca^2+^ channels such as the endoplasmic reticulum, Golgi apparatus, and mitochondria [67]. Neuroactive ligand–receptor interaction can be related to neuronal activity, and it can enhance or inhibit neuronal activity, is related to cardiac function, regulates apoptosis, and may be related to activity ability and pigmentation [68]. The hedgehog signaling pathway is one of the key regulators of animal development, and the lack of this pathway can cause changes in mouse bones and muscles and affect metabolic capacity [69]. The exercise capacity of an animal affects the proportion of its muscle fibers, which, in turn, leads to changes in IMF, which affects its lean muscle percentage, resulting in changes in marbling [70]. In the GO functional enrichment analysis, these genes were enriched in pathways such as the positive regulation of protein ubiquitination, guanyl-nucleotide exchange factor activity, and protein homodimerization activity. The remaining pathways are mostly related to neural guidance and metabolism, which suggests that our intuitive impression of flesh color is not only guided by flesh color-related genes but is also the result of the coordinated expression of multiple genes.

## 4. Conclusions

In this study, the IMF and flesh color of 1518 DLY commercial pigs were analyzed in two time periods (45 min and 12 h) for GWAS, and we found SNPs significantly associated with flesh color scores in a haplotype block spanning 79 Kb on SSC2. The heritability of the traits 45 mFC, 12 hFC, 45 mIMF, and 12 hIMF were 0.112, 0.217, 0.139, and 0.178, respectively. Most of the genes related to flesh color scores were located in the 123–127 Mb range. Other genes related to IMF are located on SSC2, SSC4, and SSC15. A total of seven candidate genes (*SNCAIP*, *PRR16*, *ST3GAL4*, *GALR1*, *ABLIM3*, *DPH5,* and *DOCK10*) were mapped for flesh color and IMF score, and we expect that seven genes related to meat quality traits can provide reference for the subsequent genetic improvement of meat quality.

## Figures and Tables

**Figure 1 genes-13-02131-f001:**
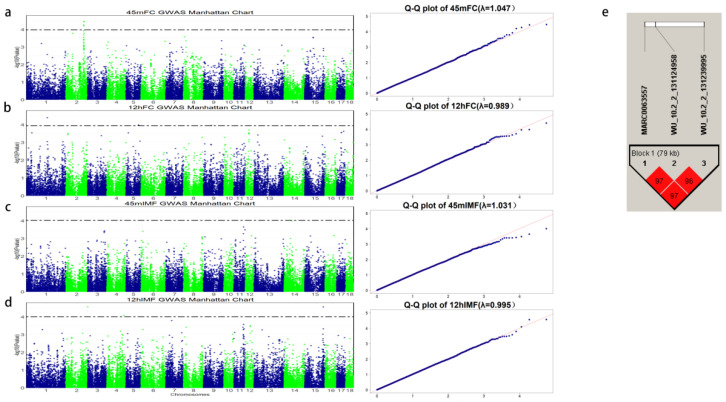
Manhattan plots of the single-locus GWAS and linkage disequilibrium. (**a**) 45-min GWAS analysis results of flesh color score. (**b**) IMF score 45-min GWAS analysis results. (**c**) 12-h GWAS analysis results of meat color score. (**d**) IMF score 12-h GWAS analysis results. (**e**) Haplotype blocks of top SNPs located on SSC2 for meat color traits. A haplotype block of 79 kb harbors the top SNP (shown in red and highlighted with black triangles). The x-axis represents chromosomes, the y-axis represents -log10 (*p*-value), and the dashed line represents the FDR correction threshold. λ is the genome expansion coefficient, the abscissa is the quantile of the expected population distribution, and the ordinate is the quantile of the sample empirical distribution.

**Table 1 genes-13-02131-t001:** Statistics of the phenotypic traits.

Phenotype ^1^	Traits ^2^	Mean (SD) ^3^	CV ^4^	*h*^2^(se) ^5^	Phenotypic Correlation ^6^		*P^7^*
Flesh color	45 mFC	3.650 ± 1.006	0.276	0.112 ± 0.032	0.343	0.135 (45 m) 0.293 (12 h)	2.2 × 10^−16^
	12 hFC	2.193 ± 0.925	0.422	0.217 ± 0.042			1.4 × 10^−7^
Intramuscular fat	45 mIMF	1.746 ± 0.717	0.411	0.139 ± 0.041	0.238		2.2 × 10^−16^
	12 hIMF	1.515 ± 0.638	0.421	0.178 ± 0.041			2.2 × 10^−16^

^1^ Flesh color and IMF phenotypic traits; ^2^ each trait is divided into different time points for statistics; ^3^ mean ± standard deviation; ^4^ coefficient of variation; ^5^ heritability ± standard error; ^6^ phenotypic correlations of the four traits (45 m is the correlation between flesh color and IMF at 45 min; 12 h is the correlation at 12 h); ^7^ phenotype-related *p*-values.

**Table 2 genes-13-02131-t002:** The four traits’ SNP locations and gene interval information.

Phenotype ^1^	SSC ^2^	SNP	Position (bp) ^3^	*P*-Value ^4^	Gene	Range ^5^
45 mFC	2	WU_10.2_2_131124958	125,961,057	3.527 × 10^−5^	*SNCAIP*	−4195
45 mFC	2	ALGA0015705	124,966,025	3.695 × 10^−5^	*ENSSSCG00000041876*	within
45 mFC	2	DRGA0003514	123,998,374	5.468 × 10^−5^	*PRR16*	+201,134
45 mFC	2	WU_10.2_2_130487175	125,006,692	6.348 × 10^−5^	*ENSSSCG00000041876*	within
12 hFC	1	ASGA0004802	147,198,001	4.044 × 10^−5^	*GALR1*	+245,820
12 hFC	5	ALGA0033448	88,285,420	1.042 × 10^−4^	*ENSSSCG00000000905*	+7341
12 hFC	9	WU_10.2_9_59299494	53,460,098	1.106 × 10^−4^	*ST3GAL4*	within
12 hIMF	2	H3GA0054148	150,317,009	2.728 × 10^−5^	*ABLIM3*	within
12 hIMF	4	INRA0017040	117,256,570	8.211 × 10^−5^	*DPH5*	+12,483
12 hIMF	15	WU_10.2_15_140224592	126,725,046	2.778 × 10^−5^	*DOCK10*	−13,515

^1^ 45 mFC, flesh color of 45 min; 12 hFC, flesh color of 12 h; 12 hIMF, intramuscular fat content of 12 h; ^2^ SSC, sus scrofa chromosome; ^3^ SNP position in Ensembl; ^4^
*p*-value in GWAS; ^5^ +/− the SNP located upstream/downstream of the nearest gene.

## Data Availability

The data on the SNP chips of 1815 pigs are not released for the time being because they belong to commercial companies and are not for private use. If necessary, the corresponding author can be contacted to request them.

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
