# Peer review of "Genome-Wide Association Studies for Flesh Color and Intramuscular Fat in (Duroc × Landrace × Large White) Crossbred Commercial Pigs"

_genes, 2022, doi:10.3390/genes13112131_

Round 1

Reviewer 1 Report

This is an interesting manuscript focused to detect SNPs associated with flesh color and intramuscular fat at 45 minutes and 12 hours (45mFC, 45hIMF, 12hFC, 12hIMF) in crossbred commercial pigs using GWAS analyses. The methodology is well described, but some additional results appear to be needed to enhance the discussion.

I suggest considering next minor comments to improve the manuscript:

1)    Abstract: According to guidelines described in “Instructions for authors”, this section should include a brief description of the main methods or treatments applied.

2)    Introduction: The objective does not match with the one described in the Introduction.

3)    Results: I suggest including Manhattan plots for GWAS analyses, as well as gene ontology graphs, in order to show the results more clearly.

Minor grammar comments:

-       Line 28= Insert the words “after slaughter” before the bracket.

-       Line 28= Replace the word “45hIMF” by “45mIMF”.

-       Line 29= Remove the word “respectively”.

-       Line 30= Replace the semicolon by a period sign.

-       Line 31= Replace the semicolon by a period sign.

-       Line 31= Remove the word “four”.

-       Line 32= Replace “is” by “was”.

-       Line 32= Remove the period after “0.178”.

-       Line 32= It is not clear with which traits the 7 and 3 SNPs were associated.

-       Line 37= Replace “pigmentation, skeletal” by “pigmentation and skeletal”.

-       Line 47= Replace “etc” by “etc.”

-       Line 48= Replace “Which” by “It”.

-       Lines 70-72= This sentence is confuse, please rewrite.

-       Line 72= Separate the bracket from the period sign.

-       Lines 93-95= This sentence is confuse, please rewrite.

-       Lines 101-102= Move “Genotype” sentence 5 lines forward.

-       Line 128= Remove the semicolon.

-       Line 139= Replace “is” by “was”.

-       Line 140= Replace “According to the Gebriel…” by “According to Gabriel…”.

-       Line 141= Insert the word “was” after “search for genes”.

-       Line 150= Remove the word “And”.

-       Line 152= Replace “are” by “were”.

-       Lines 153-157= Very long paragraph, please separate in two shorter paragraphs.

-       Line 158= Replace “is” by “was”.

-       Line 168= Replace “may cause” by “may be caused”.

-       Line 183= Insert the word “respectively” after the scores.

-       Lines 214-218= This sentence is confuse, please rewrite.

-       Line 224= Replace “in12hFC” by “in 12hFC”.

-       Line 228= Replace the semicolon by a period sign.

-       Line 255= Remove “S. M”.

-       Line 256= Replace the comma by a semicolon.

Author Response

Reviewer #1

This is an interesting manuscript focused to detect SNPs associated with flesh color and intramuscular fat at 45 minutes and 12 hours (45mFC, 45hIMF, 12hFC, 12hIMF) in crossbred commercial pigs using GWAS analyses. The methodology is well described, but some additional results appear to be needed to enhance the discussion.

I suggest considering next minor comments to improve the manuscript:

Response: We sincerely thank the editor and all reviewers for their valuable feedback that we have used to improve the quality of our manuscript. The reviewer comments are laid out below in normal font and specific concerns have been numbered. Our response is given in italicized font and changes/additions to the manuscript are given in the red text.

Q1: Abstract: According to guidelines described in “Instructions for authors”, this section should include a brief description of the main methods or treatments applied.

Response: Thank you for your suggestion, we added the description of the main methods, added content is "In this study, we determined the flesh color and intramuscular fat at 45 minutes and 12 hours after slaughter (45mFC, 45mIMF, 12hFC, 12hIMF) of 1,518 commercial Duroc × Landrace × Large White (DLY) pigs. We performed single nucleotide polymorphism (SNP) Genome-wide association study (GWAS) analysis with 28,066 SNPs." and "Furthermore, GO and KEGG analysis revealed that these genes are involved in the regulation of apoptosis and are implicated in functions such as pigmentation and skeletal muscle metabolism.". Please see Page 1 Lines 26-30 and Lines 36-38 in the revision

Q2: Introduction: The objective does not match with the one described in the Introduction.

Response: Thank you for this point. We changed it to "we used a 1,518 Duroc × Landrace × Large White (DLY) crossbred pigs population to analyzed the four traits (45mFC, 12hFC, 45mIMF, 12hIMF) through GWAS, combined with the actual character correlation and heritability, we expect provide a new sight for the identification of candidate genes for flesh color and IMF, molecular breeding of pigs.". Please see Page 2 Lines 81-84.

Q3: Results: I suggest including Manhattan plots for GWAS analyses, as well as gene ontology graphs, in order to show the results more clearly.

Response: Thank you for your suggestions on the results. We have redrawn the Manhattan plot (as shown in Figure 1 in the revision), and the gene ontology enrichment analysis result plot has been uploaded to supplement Figure S2.

Q4: Minor grammar comments:

Line 28 = Insert the words "after slaughter" before the bracket.

Response: The text has been modified according to the reviewer’s suggestion.

Line 28 = Replace the word "45hIMF" by "45mIMF".

Response: The text has been modified according to the reviewer’s suggestion.

Line 29 = Remove the word "respectively".

Response: The text has been modified according to the reviewer’s suggestion.

Line 30 = Replace the semicolon by a period sign.

Response: The text has been modified according to the reviewer’s suggestion.

Line 31 = Replace the semicolon by a period sign.

Response: The text has been modified according to the reviewer’s suggestion.

Line 31 = Remove the word "four".

Response: The text has been modified according to the reviewer’s suggestion.

Line 32 = Replace "is" by "was".

Response: The text has been modified according to the reviewer’s suggestion.

Line 32 = Remove the period after "0.178".

Response: The text has been modified according to the reviewer’s suggestion.

Line 32 = It is not clear with which traits the 7 and 3 SNPs were associated.

Response: Thank you for your suggestion. We have changed "identified 7 and 3 SNPs significantly associated with..." to "identified 7 SNPs for Flesh color and 3 SNPs for IMF." Please see Page 1 Lines 32.

Line 37 = Replace "pigmentation, skeletal" by "pigmentation and skeletal".

Response: The text has been modified according to the reviewer’s suggestion.

Line 47 = Replace "etc" by "etc.".

Response: The text has been modified according to the reviewer’s suggestion.

Line 48 = Replace "Which" by "It".

Response: The text has been modified according to the reviewer’s suggestion.

Lines 70-72 = This sentence is confuse, please rewrite.

Response: Thank you for your suggestion. We have changed "The results of previous studies are either small population size or the purebred pigs selected for the population..." to "Since previous studies have probably been mainly limited to population size or purebred, resulting in large QTL spacing, and the causal genes have not been identified. In order to reveal the genetic mechanism of meat quality, researchers use genome method to conduct in depth analysis.". Please see Page 2 Lines 72-75.

Line 72 = Separate the bracket from the period sign.

Response: The text has been modified according to the reviewer’s suggestion.

Lines 93-95 = This sentence is confuse, please rewrite.

Response: Thank you for your suggestion. We have changed "Longissimus dorsi muscle for scoring between the 11th and 12th ribs at 45 minutes and 12 hours after slaughter for..." to "The longissimus dorsi muscle between the 11th and 12th ribs was taken 45 minutes and 12 hours after slaughter and scored for flesh color and IMF using the US NPPC (1991)". Please see Page 2-3 Lines 97-99.

Lines 101-102 = Move “Genotype” sentence 5 lines forward.

Response: Thank you for your suggestion. We moved "Genotyping according to the description by Ding et al." before "According to the manufacturer’s requirements...". Please see Page 3 Lines 107.

Line 128 = Remove the semicolon.

Response: The text has been modified according to the reviewer’s suggestion.

Line 139 = Replace "is" by "was".

Response: The text has been modified according to the reviewer’s suggestion.

Line 140 = Replace "According to the Gebriel..." by "According to Gabriel...".

Response: The text has been modified according to the reviewer’s suggestion.

Line 141 = Insert the word "was" after "search for genes".

Response: The text has been modified according to the reviewer’s suggestion.

Line 150 = Remove the word "And".

Response: The text has been modified according to the reviewer’s suggestion.

Line 152 = Replace "are" by "were".

Response: The text has been modified according to the reviewer’s suggestion.

Lines 153-157 = Very long paragraph, please separate in two shorter paragraphs.

Response: Thank you for your suggestion. We have changed "Inaddition, this study found that the heritability of both flesh color and IMF at 12 hours was greater than that at 45 minutes, and the gap in the heritability of flesh color was more obvious. In comparison with previous studies [45], we obtained different results due to different measurement methods and breeds, suggesting that there is considerable potential for genetic improvement in flesh color and IMF." Please see Page 4 Lines 160-164.

Line 158 = Replace "is" by "was".

Response: The text has been modified according to the reviewer’s suggestion.

Line 168 = Replace "may cause" by "may be caused".

Response: The text has been modified according to the reviewer’s suggestion.

Line 183 = Insert the word "respectively" after the scores.

Response: The text has been modified according to the reviewer’s suggestion.

Lines 214-218 = This sentence is confuse, please rewrite.

Response: Thank you for your suggestion. We have changed "We used haplotype software for LD analysis of SNPs, there is a 79 Kb haplotype block near the top SNP (Wu_10.2_2_13124958, Figure 1e).". Please see Page 6 Lines 223-224.

Line 224 = Replace "in12hFC" by "in 12hFC".

Response: The text has been modified according to the reviewer’s suggestion.

Line 228 = Replace the semicolon by a period sign.

Response: The text has been modified according to the reviewer’s suggestion.

Line 255 = Remove "S. M".

Response: The text has been modified according to the reviewer’s suggestion.

Line 256 = Replace the comma by a semicolon.

Response: The text has been modified according to the reviewer’s suggestion.

Reviewer 2 Report

Congratulation! Well Done!

Flesh color and intramuscular fat are subjective indicators in meat quality assessment, but the consumers are looking for those...so we have to take those into consideration! 

Acronim IMF have to be explain in the resume, like the first time when you used it in text.

Manhattan plots - the threshold line have to be more clear - all figure have to be clearer.

The conclusions have to be focused on results not on the no of animals of the study, materials/methods (first part of first paragraph can be removed)

Author Response

Reviewer #2

Congratulation! Well Done!

Flesh color and intramuscular fat are subjective indicators in meat quality assessment, but the consumers are looking for those..., so we have to take those into consideration!

Response: We feel great thanks for your and editor professional review work on our article. As you are concerned, there are several problems that need to be addressed. According to your nice suggestions, we have made extensive corrections to our previous draft and our response is given in italicized font and changes/additions to the manuscript are given in the red text.

Q1: Acronim IMF have to be explain in the resume, like the first time when you used it in text.

Response: Thank you for your suggestion. We made a correction in the abstract in Page 1 line 25.

Q2: Manhattan plots - the threshold line have to be more clear - all figure have to be clearer.

Response: Thank you for your suggestion. We have improved the definition of the image, and the pixel of the image is now 300 dpi (as shown in Figure 1 in the revision), so we can clearly observe the experimental results.

Q3: The conclusions have to be focused on results not on the no of animals of the study, materials/methods (first part of first paragraph can be removed)

Response: Thank you for your valuable advice, we have changed "In this study, the IMF and flesh color of 1,518 DLY commercial pigs were analyzed in two time periods (45 minutes, 12 hours) for GWAS," and "A total of 7 candidate genes (SNCAIP, PRR16, ST3GAL4, GALR1, ABLIM3, DPH5 and DOCK10)". Please see Page 8 Lines 312-313 and Lines 317-318.
